# Electrical and Structural Insights into Right Ventricular Outflow Tract Arrhythmogenesis

**DOI:** 10.3390/ijms241411795

**Published:** 2023-07-22

**Authors:** Yen-Yu Lu, Yao-Chang Chen, Yung-Kuo Lin, Shih-Ann Chen, Yi-Jen Chen

**Affiliations:** 1Division of Cardiology, Department of Internal Medicine, Sijhih Cathay General Hospital, New Taipei City 22174, Taiwan; yolu59@yahoo.com.tw; 2School of Medicine, Fu-Jen Catholic University, New Taipei City 24257, Taiwan; 3Department of Biomedical Engineering, National Defense Medical Center, Taipei 11490, Taiwan; yaochang.chen@gmail.com; 4Division of Cardiovascular Medicine, Department of Internal Medicine, Wan Fang Hospital, Taipei Medical University, Taipei 11696, Taiwan; yklin213@yahoo.com.tw; 5Cardiovacular Research Center, Wan Fang Hospital, Taipei Medical University, Taipei 11696, Taiwan; 6Heart Rhythm Center, Division of Cardiology, Department of Medicine, Taipei Veterans General Hospital, Taipei 11217, Taiwan; epsachen@ms41.hinet.net; 7Cardiovascular Center, Taichung Veterans General Hospital, Taichung 40705, Taiwan; 8Department of Post-Baccalaureate Medicine, College of Medicine, National Chung Hsing University, Taichung 40227, Taiwan; 9Graduate Institute of Clinical Medicine, College of Medicine, Taipei Medical University, Taipei 11696, Taiwan

**Keywords:** right ventricular outflow tract, ventricular arrhythmia

## Abstract

The right ventricular outflow tract (RVOT) is the major origin of ventricular arrhythmias, including premature ventricular contractions, idiopathic ventricular arrhythmias, Brugada syndrome, torsade de pointes, long QT syndrome, and arrhythmogenic right ventricular cardiomyopathy. The RVOT has distinct developmental origins and cellular characteristics and a complex myocardial architecture with high shear wall stress, which may lead to its high vulnerability to arrhythmogenesis. RVOT myocytes are vulnerable to intracellular sodium and calcium overload due to calcium handling protein modulation, enhanced CaMKII activity, ryanodine receptor phosphorylation, and a higher cAMP level activated by predisposing factors or pathological conditions. A reduction in *Cx43* and *Scn5a* expression may lead to electrical uncoupling in RVOT. The purpose of this review is to update the current understanding of the cellular and molecular mechanisms of RVOT arrhythmogenesis.

## 1. Introduction

The right ventricular outflow tract (RVOT) tachycardia is a form of monomorphic or polymorphic ventricular tachycardia originating from the RVOT and accounts for the majority (80%) of outflow tract ventricular arrhythmias. The RVOT is a dominant site of origin of premature ventricular contractions and ventricular tachyarrhythmias that are often observed in patients without structural heart diseases, including idiopathic ventricular arrhythmias, Brugada syndrome, torsade de pointes, and long QT syndrome. The RVOT is also one of the origins of malignant ventricular tachycardia caused by structural heart disease, such as arrhythmogenic RV cardiomyopathy (ARVC) [1,2,3,4]. It typically presents in young patients and its incidence is increasing notably [5]. The majority of patients of RVOT arrhythmias are initially diagnosed between the age of 30 and 50 years [6]. RVOT arrhythmias can originate from different sites of the RVOT, including the rightward (free wall), anterior, leftward and posterior (septal) parts, but carry common ECG characteristics. General features of ventricular tachycardia from RVOT are left bundle branch block morphology, an inferior axis, and an rS complex in V1 and R complex in V6. However, the intimate and complex anatomy of the outflow tract limits the predictive value of ECG criteria alone to distinguish RVOT tachycardia from supraventricular tachycardia with left bundle branch block or ventricular tachycardia originating from areas closed to the RVOT, such as the left ventricular outflow tract and the aortic sinus cusp, etc. For clinical purposes, ECG characteristics to differentiate these arrhythmias from RVOT tachycardia have been studied [7,8,9]. Except for ECG, late potentials on the SAECG were present in 78% of the patients with ARVC/D but were not present in any patient with RVOT tachycardia, and ventricular tachycardia in ARVC does not terminate with adenosine [10]. Buxton et al. published the first description of RVOT tachycardia in 1983, and a quarter of patients were asymptomatic [11]. RVOT arrhythmias are usually considered benign and may result in recurrent symptoms of palpitation and dizziness. A small subset of patients with RVOT arrhythmias have tachycardias that are extremely rapid or result in loss of consciousness, and very rarely, aborted sudden death [12]. Triggering beats from the RVOT have a crucial role in initiating ventricular fibrillation associated with Brugada syndrome, as confirmed by the elimination of triggers and malignant arrhythmias after ablation [13]. Emerging laboratory evidence has found that rabbit RVOT myocytes contain distinctive electrophysiological characteristics, which may lead to their high vulnerability to arrhythmogenesis [14]. Moreover, structural heart diseases, the dysfunction of autonomic nervous activity, and sex hormones have a great impact on RVOT arrhythmogenesis due to electrical and structural remodeling. In this review, we update the current understanding of the electrical and structural insights into RVOT arrhythmogenesis, and discuss the impact of predisposing factors and pathological conditions on RVOT arrhythmogenesis.

## 2. Predisposing Factors of RV Arrhythmogenesis

### 2.1. Anatomical and Dynamic Perceptions on RVOT Arrhythmogenesis 

The RVOT refers to the region of the right ventricle (RV) between the supraventricular crest and the pulmonary valve and antero-cephalad to the left ventricle, resulting in a characteristic “crossover” relationship between the right and left ventricular outflows. The RVOT is a thin, smooth-walled, tubular structure, and comprises a complex three-dimensional network of myocardial fibers arranged circumferentially in the sub-epicardium and longitudinally in the sub-endocardium [15]. Muscular distribution of the RVOT is relatively thin in the free wall and immediate subpulmonary valve portions and thicker in the proximal posterior portion, where it is directly opposite the left ventricular outflow tract and the anterior interventricular septum. Patients with idiopathic RVOT tachycardia were found to have structural anomalies in magnetic resonance imaging, with the RV free wall being the most often aberrant origin [16]. The RVOT would seem more susceptible to dynamic muscular obstruction given the presence of a circumferential, muscular, contractile infundibulum in contrast to the partly fibrous, non-contractile left ventricular outflow tract. The presence of an RVOT gradient was more common in younger, male patients [17]. In a healthy athletic male heart, the RVOT is exposed to high shear stress, which is higher at the proximity of the RVOT free wall compared to the septal RVOT and other RV regions [18]. A relative increase in wall shear stresses in the RVOT, particularly in the RVOT free wall, may contribute to adverse remodeling, scar formation, and arrhythmia. At higher cardiac-output levels, the RVOT wall shear stress increases in a somewhat linear manner. The identified regions of elevated hemodynamic stress at resting and exercise conditions correlate with the common site of RVOT tachycardia in athletes and diseased regions in patients with ARVC [18].

### 2.2. Embryonic Development Relates to RVOT Arrhythmogenesis

The RVOT and left ventricular outflow tract have a common origin. The subpulmonary myocardium, which corresponds to the adult RVOT, is formed during the fetal period when a large inferior portion of the muscular embryonic outflow tract is incorporated into the RV free wall, and shows different gene expression from the superior part of the embryonic outflow tract [19]. The inferior part of the muscular embryonic outflow tract expresses *Sema3C* and the subpulmonary myocardium is specifically affected and possibly largely absent in *Tbx1* mutant mice [19,20]. During development, *Hey2* is expressed in the ventricular working myocardium in a transmural gradient and represses the expression of *Scn5a* and is absent from the outflow tract. *Hey2* is expressed in the RVOT of adult mouse heart. The gene variants may influence the expression of *HEY2*, leading to reduced *SCN5A* expression and a reduced sodium (Na^+^) current, thereby mimicking a loss of function mutation in *SCN5A* as found in a minority of patients with Brugada syndrome [21]. Embryological studies have shown that the RVOT and RV myocytes have different origins, and the conducting phenotype of the RVOT differs from that of the RV free wall and of the left ventricle. In the developing embryonic heart, the myocardial outflow tract presents “transitional zones” with slow conducting properties. Working myocardial genes involved in fast conduction, for example, *Cx43* and *Scn5a*, are less expressed in the adult mouse RVOT [22]. The slow conducting phenotype of the embryonic RVOT in hearts with the Brugada syndrome-linked *Scn5a* mutation is maintained in the adult heart, resulting in a lower conduction reserve in the RVOT [22]. Additionally, the hyperpolarization-activated cyclic nucleotide-gated channel 4 (HCN4) has been found to be expressed in the RVOT of the developing heart, and the re-expression of an embryonic phenotype or embryonic remnants of tissue may provide the arrhythmogenic potential of this area [23]. These findings may contribute to the increased propensity for arrhythmias to arise there.

### 2.3. Histopathological Perceptions on RVOT Arrhythmogenesis

The RVOT tissue structure is interspersed with the heterogeneous distribution of fibrous and fat tissue [24,25], and comprises a complex myocardial architecture [15]. The musculature sleeves of RVOT with fibrous and fatty connective tissue possess a unique arrhythmogenic substrate or triggered activity that results in ventricular arrhythmias [26]. Idiopathic RVOT tachycardia often arise from myocardial sleeves and are catecholamine-sensitive, suggesting that automaticity or triggered activity may contribute to RVOT arrhythmogenesis [27]. Myocardial fibrosis has been suggested by abnormal, low-voltage, fractionated electrograms localized to the RVOT at the epicardium [28]. Among patients with Brugada syndrome, interstitial fibrosis of the RVOT with conduction delay and the epicardial–endocardial gradient of collagen deposition has been reported [29,30]. Connexin 43 is poorly expressed in rabbit RVOT with electrical uncoupling, and may therefore be important in the pathogenesis of Brugada syndrome [31]. Intramural fibrosis is an exacerbating factor for the appearance of RVOT arrhythmias. A greater density of intramural clefts being associated with a reduced Na^+^ channel function was found in mouse RVOT myocytes, resulting in an increased vulnerability to conduction abnormalities [32]. Given the complex anatomic makeup and the presence of the endocardial Purkinje network [33], the RVOT could be predisposed to reentry [34]. Moreover, fat not only covers the RVOT epicardial surface but also infiltrates the underlying myocardium (i.e., fatty infiltration) and localizes around vessels and nerves or separate strands of myocardial fibers [25]. Intramyocardial fatty infiltration increased most extensively in the RVOT in human hearts with ARVC [35].

### 2.4. Cellular Electrophysiology Perceptions on RVOT Arrhythmogenesis

Triggered activity is considered to be the mechanism underlying adenosine-sensitive RVOT tachycardia due to catecholamine-induced delayed afterdepolarization (DAD). Catecholamine stimulation causes increases in intracellular cyclic adenosine monophosphate (cAMP), L-type calcium (Ca^2+^) current (I_Ca-L_), and a spontaneous oscillatory release of Ca^2+^ from the sarcoplasmic reticulum that activates a transient inward current, giving rise to a DAD [36]. The functional dynamics of action potentials (APs) and repolarization heterogeneity in the RVOT are believed to be major contributors to Brugada-type electrocardiographic changes [37,38]. In patients with Brugada syndrome, there was a negative voltage gradient between the epicardium and endocardium of the RVOT, with significantly prolonged activation time in the endocardium and the activation-recovery interval in the epicardium, resulting in a larger area of ST-segment elevation >2 mV and T-wave inversions [39]. Compared to RV apex myocytes, RVOT myocytes had a wider range of AP duration (APD) in a rabbit model [40]. Longer APD in RVOT myocytes may contribute to ventricular arrhythmias arising mainly from RVOT in patients with long QT syndrome [41]. Spatial heterogeneity exists in repolarization and APD between epicardial and endocardial RVOT and the RV apical region under physiological conditions [34]. Arrhythmia in the RVOT is associated with positive spatiotemporal autocorrelation between the epicardial–endocardial arrhythmic wave fronts and reentrant rotors [34]. Moreover, regional and tissue transcript signatures of ion channel genes also play a role in RVOT arrhythmogenesis. Higher expression of the Na^+^ channel was found at the basal endocardium of RVOT in non-diseased human hearts [42]. As shown in Figure 1, our previous study identified, for the first time, distinctive electrophysiological characteristics in RVOT myocytes with a longer APD compared to that in the RV apex [14]. The late Na^+^ current (I_Na-Late_) flows during the late-phase of AP to prolong APD and plays an important role in the genesis of cardiac arrhythmias as a result of enhanced triggered activity. A single *Scn5a* insertion mutation produces an early Na^+^ channel closure but augments the I_Na-Late_, which may present in patients with features of both Brugada syndrome and long QT-3 syndrome [43]. As the transient outward potassium (K^+^) current (I_to_) plays a pivotal role in the genesis of Brugada syndrome, the larger I_to_ in RVOT myocytes may mediate a more prominent phase 1 notch in AP morphology [14], resulting in a high arrhythmogenesis in those patients [37]. Liang S. et al. showed a wider range of APDs in rabbit RVOT myocytes than in rabbit RV myocytes. The I_Ca-L_ was obviously larger in the long-APD-RVOT myocytes, which is correlated with the occurrence of early afterdepolarization (EAD) [40]. In our previous study, rabbit RVOT myocytes had a reduced I_Ca-L_ that relates to Ca^2+^ overload_._ Moreover, the smaller rapid component of the delayed rectifier K^+^ current (I_Kr_) may produce a delay in phase 3 repolarization of the AP and the I_Na-Late_ flows during the late phase of the AP in RVOT myocytes, resulting in their longer APD. Both reduced I_Kr_ and increased I_Na-Late_ are known to cause long QT syndrome, a clinical condition associated with an increased risk for torsades de pointes-type of ventricular tachycardia.

### 2.5. Ca^2+^ Homeostasis Perceptions on RVOT Arrhythmogenesis 

An imbalance in Ca^2+^ homeostasis in a cardiomyocyte can lead to electrical disturbance. Normally, depolarization opens voltage-gated Na^+^ channels that lead to large Ca^2+^ influxes after opening the L-type Ca^2+^ channels and some Ca^2+^ through the T-type Ca^2+^ channels and the reverse mode Na^+^-Ca^2+^ exchanger (NCX), leading to an increase in the intracellular Ca^2+^ concentration. The increased Ca^2+^ concentration triggers Ca^2+^ to release from the sarcoplasmic reticulum into the cytoplasm via the ryanodine receptor, a process known as Ca^2+^-induced Ca^2+^ release [44]. The increased cytosolic Ca^2+^ level then decreases through Ca^2+^ reuptake into the sarcoplasmic reticulum by the sarco/endoplasmic reticulum Ca^2+^-ATPase and Ca^2+^ extrusion by the sarcolemmal NCX and plasma membrane Ca^2+^-ATPase [45], given that the level of resting intracellular Ca^2+^ represents the balance between Ca influx and extrusion. In some pathological conditions, Ca^2+^ overload is found when intracellular Ca^2+^ removal becomes impaired with increased sarcoplasmic reticulum and intracellular Ca^2+^ concentrations, leading to the increased Ca^2+^ leak by ryanodine receptors [46]. DAD and EAD by spontaneous Ca^2+^ release from the sarcoplasmic reticulum are observed under the conditions of Ca^2+^ overload, induce triggered activity, resulting in ventricular tachycardia and ventricular fibrillation [47]. Late-phase 3 EAD combines the properties of both EAD and DAD and induces triggered activity in isolated canine Purkinje fibers [48]. Late-phase 3 EAD occurs under Ca^2+^ overload and the lengthening of the Ca^2+^ transient duration and is associated with a shortening in APDs [49,50]. Late-phase 3 EAD can occur immediately after the conversion to sinus rhythm from ventricular tachycardia or fibrillation [50]. A higher expression of Ca^2+^-handling genes was found at the basal epicardium of human RVOT [42]. Ca^2+^ overload is known to play a major role in RVOT arrhythmogenesis. The potential Ca^2+^ dysregulation in RVOT myocytes suggests that the RVOT may have a propensity for ventricular arrhythmias. RVOT myocytes had a faster decline in the intracellular Ca^2+^ transient, suggesting that sarcoplasmic reticulum Ca^2+^ uptake is faster in RVOT myocytes. A higher sarcoplasmic reticulum Ca^2+^ load may lead to a spontaneous Ca^2+^ wave and produce the triggered activity of DAD [51]. The NCX plays a major role in removing intracellular Ca^2+^ and decreasing sarcoplasmic reticulum Ca^2+^ content. The larger I_Na-Late_ in RVOT myocytes may alter the rate of Na^+^ extrusion/Ca^2+^ entry and lead to intracellular Ca^2+^ overload in different species [52], resulting in the genesis of EAD. Ouabain inhibits the Na^+^-K^+^ pump and increases I_Na-Late_ by Ca^2+^ calmodulin-dependent protein kinase II (CaMKII) that causes Ca^2+^ overload in ventricular myocytes [53,54]. Ouabain infusions have triggered DAD and induced sustained ventricular tachycardia during in vivo and in vitro studies [14,51]. Either the CaMKII inhibitor (KN-93) or an I_Na-Late_ inhibitor (ranolazine) can terminate ouabain-induced ventricular arrhythmias in rabbit RVOT but not in rabbit RV apex. Accordingly, CaMKII and I_Na-Late_ inhibition on RVOT tissues ameliorates ouabain-induced ventricular tachycardia with Na^+^ and Ca^2+^ overload (Figure 2) [14].

### 2.6. Excitation–Contraction Coupling on RVOT Arrhythmogenesis

Cardiac excitation–contraction coupling links cardiac mechanics and electrophysiology with the integration of APD and muscle contractility [55,56]. In the steady state without stimulation, the level of resting intracellular Ca^2+^ is controlled entirely by the surface membrane because there is no net Ca^2+^ flux across the membrane of the sarcoplasmic reticulum [57]. When heart beat is increasing, the amplitude of the systolic Ca^2+^ transient decreases and is accompanied by an increase in diastolic intracellular Ca^2+^ [58]. Rapid activation rates mean that the previous Ca^2+^ transient has insufficient time to decay and cause the increased intracellular Na^+^ activated reverse mode NCX to increase the intracellular Ca^2+^ load. Upon the return to a normal rate after rapid activation, augmented levels of sarcoplasmic reticulum Ca^2+^ loading and release stimulate the extrusion of Ca^2+^ through the forward mode NCX, resulting in a transient period of hypercontractility, APD prolongation and EAD development [52,55]. RVOT arrhythmogenesis resulting from the altered excitation–contraction coupling of RVOT tissue substrate has been reported in patients with Brugada syndrome [59]. Our previous studies demonstrated that a concurrent prolonged post-pacing phase 2 of APD and an enhanced contractility may trigger ventricular tachycardia, resulting from persistent inward Ca^2+^ currents in the short QT tissue model in rabbit RVOT [60]. In contrast, RVOT tachycardia is less likely to occur in those with post-pacing trivially enhanced contractility or APD prolongation. In the drug-induced long QT syndrome model, an increase in contractility, rather than an increase in APD, is critical for the genesis of ventricular tachyarrhythmias at the RVOT [61]. The occurrence of post-pacing-induced ventricular arrhythmias could be ameliorated by an NCX inhibitor, suggesting that NCX plays a role in disruption of the intracellular Ca^2+^ homeostasis in the cardiac excitation–contraction coupling as a crucial mechanism in triggering ventricular arrhythmias [62].

### 2.7. Role of Autonomic Nervous Activity in RVOT Arrhythmogenesis

Autonomic-nervous-system abnormality may provide a pathophysiological basis for inherited sudden cardiac arrest syndrome [63]. The autonomic dysfunction associated with sympathovagal imbalances has been suggested to play a critical role in the generation of RVOT arrhythmia [64]. Sympathetic fibers of the ventromedial cardiac nerve and branches of the ventrolateral cardiac nerve innervate the myocardium with the proximal pulmonary artery and the RVOT [65]. The sympathetic nerve innervations were found to increase in canine RVOT, especially at the site of arrhythmogenic origin [66]. Moreover, the reduced reuptake of norepinephrine into the nerve terminal or its increased release into the synaptic cleft leads to an increase in local norepinephrine concentrations in the synaptic cleft and the stimulation of postsynaptic beta-adrenoceptors, leading to increased cAMP concentrations. The increase in cAMP will produce Ca^2+^ overload and trigger RVOT tachycardia [67]. The induced sustained RVOT tachycardia is adenosine-sensitive and verapamil-sensitive and is caused by an automatic mechanism [4]. The response of the sustained RVOT tachycardia to catecholamine is usually attenuated by beta-blockers, Ca^2+^ channel blockers and adenosine [68], which provides indirect evidence of a sympathetic tone in the arrhythmogenesis of RVOT arrhythmia. Accordingly, sympathetic hyperactivity could influence the initiation and perpetuation of RVOT tachycardia in the tachypacing-induced ventricular arrhythmias. Recently, Aras et al. reported that ventricular cholinergic innervation crossed the human RVOT transmural wall, and cholinergic stimulation counteracted the sympathetic effects of adrenergic stimulation on the change in APD and the frequency of premature ventricular contractions [34].

### 2.8. Sex Hormones

Gender differences have been observed in the epidemiology, pathogenesis and clinical presentation of various types of idiopathic RVOT tachycardia [69,70]. Women have RVOT tachycardia initiation with recognized states of hormonal flux. Men more commonly have RVOT initiated by exercise or stress [70]. Idiopathic RVOT tachycardia, long QT syndrome and drug-related torsade de pointes are more common in women than in men, while Brugada syndrome is more common in men. Male patients with ARVC have a higher mortality rate and sudden death rate than in female patients with the same mutation [71]. Brugada syndrome is common in men than in women and a higher mortality rate has been documented in male patients with ARVC [69]. The shortening of the corrected QT interval during puberty in males implies that androgens (specifically testosterone) also contribute to gender differences in the corrected QT interval [72]. Androgen receptor ablation has been shown to regulate electrophysiology and enhance tachyarrhythmia [73]. Our previous study found that androgen-receptor knockout mice had lower levels of serum testosterone, and longer corrected QT interval, with a longer APD in RVOT myocytes [74]. RVOT tissues from androgen-receptor knockout mice had a higher incidence of EADs and faster burst firing. RVOT myocytes from androgen-receptor knockout mice have an increased CaMKII and ryanodine receptor phosphorylation, which increases spontaneous Ca^2+^ release and subsequent ventricular arrhythmias. Accordingly, sex hormones change RVOT electrophysiology and Ca^2+^ homeostasis, with increased ventricular arrhythmogenesis.

### 2.9. Mutations in Various Signaling Pathways Leading to RVOT Arrhythmogenesis

Genetic defects play an important role in the genesis of inherited RVOT arrhythmogenesis and have been widely reviewed [75,76]. Inherited genetic channelopathy is a significant cause of sudden death in children and adolescents, such as ARVC, and family history and genetic screening may help identify asymptomatic carriers and are essential for early diagnosis in RVOT tachycardia [77,78]. In addition to inherited genetic channelopathy, rare variants in gap junction and transcription factor genes have been associated with RVOT arrhythmogenesis (Table 1).

## 3. Pathological Conditions for RVOT Arrhythmogenesis 

Pathological precipitating factors, resulting in acquired channelopathy, are vital to trigger the genesis of RVOT tachyarrhythmia due to electrical and structural remodeling (Figure 3).

### 3.1. Obesity/Epicardial Adipose Tissue (EAT) Accumulation in the RVOT

Overweight and obesity are important risk factors for a wide range of chronic diseases, including cardiovascular diseases, type 2 diabetes, and several types of cancer, as well as all-cause mortality [87]. Obesity has been documented to be an independent risk factor for sudden death and other cardiovascular mortality [88]. Obesity is associated with an increasing EAT amount or EAT phenotype modulation [89], and induces electrical and structural remodeling with increasing arrhythmogenic potential [90]. The prevalence of premature ventricular contractions is 30 times higher in obese patients with eccentric left ventricular hypertrophy compared with lean subjects [91]. EAT and the associated increased fibrosis could anatomically disrupt the cell-to-cell electrical conduction of cardiomyocytes, which might result in slow conduction, facilitating arrhythmogenesis. Obesity and overweight slightly increased the RVOT diameter [92]. RVOT, commonly covered with EAT, is vital for ventricular arrhythmia genesis [93]. The fat accumulation on the RVOT epicardial surface could be associated with a higher density and frequency and more fractionations of wave fronts in the epicardium compared with the endocardium [34]. Our previous study showed that EAT-connected rabbit RVOT myocytes had longer APD and an increased resting membrane potential (RMP), suggesting that EAT may change RVOT electrophysiological characteristics [94]. The decreased I_Ca-L_ might be responsible for the long APD in EAT-connected rabbit RVOT. The inward rectifier K^+^ current was more depressed in EAT-connected rabbit RVOT myocytes, which might lead to a less-negative RMP and make EAT-connected rabbit RVOT myocytes more excitable, leading to high arrhythmogenesis. These findings are in line with clinical observations in patients with Brugada syndrome [2]. Moreover, EAT-connected rabbit RVOT myocytes had larger I_to_. I_to_ exists in large amounts at the epicardial site of the RV, and is suggested to enhance phase 2 reentry like in the model of Brugada syndrome, as suggested in the repolarization disorder hypothesis [38].

### 3.2. Heart Failure (HF)

HF is a clinical syndrome with multiple etiologies leading to cardiac function impairment. It is associated with significant morbidity and mortality. The most common cause of RV dysfunction is chronic left-sided HF [95]. A previous study found that there is less RVOT fractional shortening in HF patients than healthy controls [96]. HF may have an impact on RVOT electrophysiological characteristics. The ligation of the left anterior descending coronary artery has been widely used to establish the model of HF [97]. Human HF increases the RVOT apex-to-base APD gradient, and regions of long and short APDs cause non-uniform wave propagation, which may initiate ventricular arrhythmias until unidirectional conduction block and reentry [98]. HF increases I_Na-Late_ in rabbit RVOT myocytes that allow the sustained entry of Na^+^ into the myocardial cells throughout systole, which prolongs APD and increases the risk of arrhythmogenesis [99]. HF increases NCX in rabbit RVOT myocytes [99], and upregulated NCX creates conditions that facilitate arrhythmia formation through DAD in a spontaneous Ca^2+^ wave that elevates intracellular Ca^2+^ levels during the diastolic phase [100]. The increased NCX in HF RVOT myocytes may arise from the increased I_Na-Late_ levels and compensates for the decrease in Ca^2+^ entry through I_Ca-L_ during depolarization [101]. The I_to_ and I_Kr_ reduction in HF RVOT myocytes are accompanied by APD prolongation [99]. Dysregulated Na^+^ and K^+^ currents have been demonstrated to play an important role in ventricular arrhythmogenesis in HF [99].

### 3.3. Chronic Kidney Disease (CKD)

CKD is a major burden in patients with cardiovascular diseases, and it has been demonstrated that even mild forms of CKD may be associated with cardiovascular morbidity and mortality [102]. Sudden death is common in CKD and accounts for approximately one-quarter of deaths in patients undergoing dialysis [103]. CKD induces adverse ventricular remodeling that provides the substrate for ventricular arrhythmia [104]. In the ARIC study, patients with CKD had a significant prevalence of non-sustained ventricular tachycardia (30.2%), and >90% of patients had ectopy according to 2-week cardiac monitor recording [105]. Rautavaara et al. monitored CKD patients with an implanted device for 34 months, and 23% of patients had ventricular tachyarrhythmias [106]. Recently, the RV function has been particularly highlighted for dialyzed patients [107]. Hemodialysis is associated with an increased risk of pulmonary hypertension, which causes RV structure abnormalities and dysfunction and is recognized as a predictor of mortality in these patients [108]. Comparative research by Schleberger et al. revealed that the incidence of CKD was 14% in patients undergoing catheter ablation for outflow tract arrhythmias in the absence of structural heart disease [109]. Patients with advanced CKD have been reported to have Ca^2+^ handling abnormalities, abnormal electrophysiological characteristics, and a low induction threshold of ventricular fibrillation [104]. Moreover, a significant interdialytic change in the RVOT velocity time integral was found in CKD patients on hemodialysis, resulting in a high interdialytic hemodynamic change that may contribute to an increase in RVOT wall shear stresses and subsequent RVOT remodeling [110]. Combined cephalosporin and aminoglycoside cause tubular necrosis and induce rabbit kidney injury, which have been used as a CKD animal model [111,112]. The RVOT from CKD rabbits (induced by neomycin and cefazolin) exhibit a shorter APD and a higher incidence of non-sustained ventricular tachycardia [112]. The shortened APD in CKD RVOT myocytes may have been caused by large I_to_ and I_Kr_, which commonly facilitate the repolarization of APs.

In contrast, left ventricular free wall myocytes have been shown to have a prolonged APD in CKD rats, which is caused by the downregulation of cardiac K^+^ currents [113]. The increased regional differences of APD in CKD may contribute to an enhanced APD dispersion and facilitate the genesis of the reentry circuits that result in ventricular arrhythmia. Furthermore, the long QRS duration observed in CKD rabbits may be caused by its small Na^+^ current, gap junction uncoupling and the reduced expression of connexin 43, respectively, contributing to ventricular arrhythmia [114]. A large I_to_ in CKD RVOT myocytes can be consistent with the prominent spike-and-dome morphology of the APs required for phase 2 reentry [115].

The larger I_Ca-L_ in CKD RVOT myocytes can increase Ca^2+^ entry and promote Ca^2+^ loading [112], which may cause Ca^2+^ dysregulation and CaMKII signaling pathway activation. Similar to that in HF, CKD RVOT tissues also exhibited increased levels of NCX, and the increasing Ca^2+^ leak in CKD RVOT myocytes, which leads to the genesis of triggered activity. In addition, CKD-enhanced RVOT arrhythmogenesis can be reduced by an NCX inhibitor [112]. The CKD RVOT tissues exhibited high expressions of phosphorylated CaMKII and protein kinase A and a decrease in sarcoplasmic/endoplasmic reticulum Ca^2+^ ATPase 2a (SERCA2a) expression, leading to Ca^2+^ handling abnormalities. The increase in NCX function possibly originated through CaMKII-dependent phospholamban phosphorylation.

The CKD RVOT myocytes also show an increased level of oxidative stress, which may have promoted kinase activity and CaMKII expression to cause Ca^2+^ dysregulation. The protein expression of SERCA2a and SERCA2a activity may have been inhibited by the generation of oxidative stress. The hypersympathetic innervation in the CKD RVOT suggests that the RVOT may play a role in the induction of ventricular arrhythmia in CKD through enhanced sympathetic activity. The CKD RVOT with higher oxidative stress and autonomic hyperactivity exhibited distinctive electrophysiological characteristics and Ca^2+^-handling abnormalities, which contributed to the induction of ventricular tachycardia.

## 4. Conclusions

The RVOT becomes highly arrhythmogenic because RVOT myocytes have unique electrical and Ca^2+^-handling properties that are known to cause arrhythmias. Multiple intrinsic and extrinsic factors are linked to RVOT arrhythmogenesis, which is easily induced in pathological states (Figure 4). As a result, tackling these causes and underlying diseases may provide therapeutic prospects for treating arrhythmias originating from the RVOT.

## Figures and Tables

**Figure 1 ijms-24-11795-f001:**
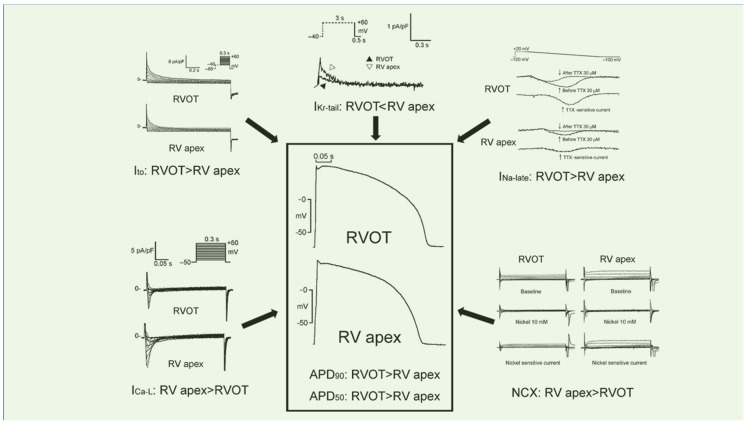
Distinctive electrophysiological features of right ventricular outflow tract (RVOT) myocytes relative to right ventricular (RV) apex myocytes. RVOT myocytes had longer action potential duration (APD) at 90% and 50% repolarization (APD_90_ and APD_50_) than RV apex myocytes. RVOT myocytes were characterized by a larger late sodium current (I_Na-Late_), transient outward potassium current (I_to_), and smaller the rapid component of the delayed rectifier potassium current (I_Kr-tail_), L-type calcium current (I_Ca-L_) and nickel-sensitive Na^+^-Ca^2+^ exchanger (NCX). The differences in the properties of the I_Na-Late_, the I_to_, the I_Kr-tail_, the I_Ca-L_ and the NCX current might contribute to the differences in APD between RVOT myocytes and RV apex myocytes. (Adapted with permission from [14]).

**Figure 2 ijms-24-11795-f002:**
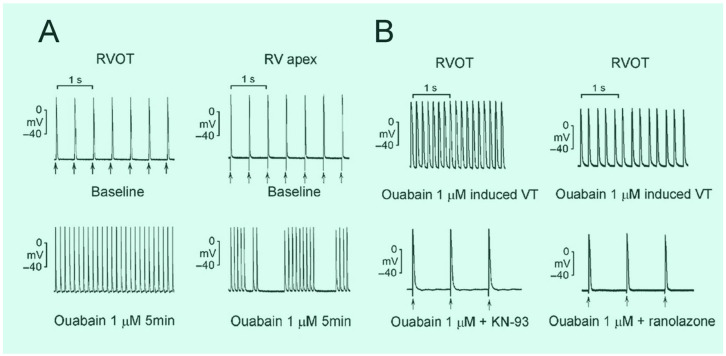
Effects of KN93 (a CaMKII inhibitor) and ranolazine (a late sodium current inhibitor) on ouabain (a cardiac glycoside)-induced ventricular tachycardia (VT). (**A**) Ouabain induces sustained VT in the right ventricular outflow tract (RVOT) and non-sustained VT in the right ventricular (RV) apex. (**B**) KN93 or ranolazine can terminate ouabain-induced sustained VT in RVOT preparations. Arrows (↑) indicate electrical stimuli. (Adapted with permission from [14]).

**Figure 3 ijms-24-11795-f003:**
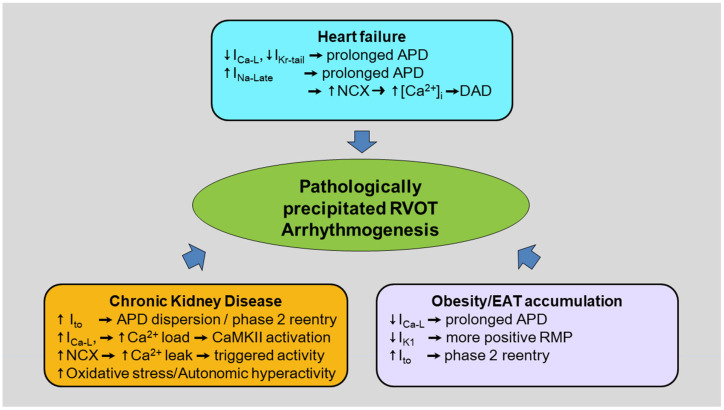
Electrical ventricular remodeling in pathological conditions, such as heart failure, chronic kidney disease and obesity/epicardial adipose tissue (EAT) accumulation, contributes to right ventricular outflow tract (RVOT) arrhythmogenesis. In heart failure, RVOT myocytes have a smaller L-type calcium (Ca^2+^) current (I_Ca-L_) and a rapid component of delayed rectifier potassium current (I_Kr-tail_) and a larger late sodium (Na^+^) current (I_Na-Late_) than RV apex myocytes, which prolongs action potential duration (APD). Moreover, an increase in intracellular Na^+^ by I_Na-Late_ may drive the Ca^2+^ inside by the reverse mode of Na^+^-Ca^2+^ exchanger current (NCX), which increases intracellular Ca^2+^ ([Ca^2+^]_i_) and induces delayed afterdepolarization (DAD). In chronic kidney disease, RVOT myocytes have a larger transient outward potassium current (I_to_), which causes APD dispersion and phase 2 reentry. CKD RVOT myocytes have a larger I_Ca-L_ and NCX than RV apex myocytes that increase Ca^2+^ load and Ca^2+^ leak, resulting in Ca^2+^ calmodulin-dependent protein kinase II (CaMKII) activation and inducing triggered activity. Moreover, chronic kidney disease increases oxidative stress and causes autonomic hyperactivity. In obesity/epicardial adipose tissue (EAT) accumulation, a smaller I_Ca-L_ and inwardly rectifying the potassium current (I_K1_) of RVOT myocytes result in a prolonged APD and a more positive resting membrane potential (RMP) than that of RV apex myocytes. RVOT myocytes have a larger I_to_ than RV apex myocytes, which causes phase 2 reentry.

**Figure 4 ijms-24-11795-f004:**
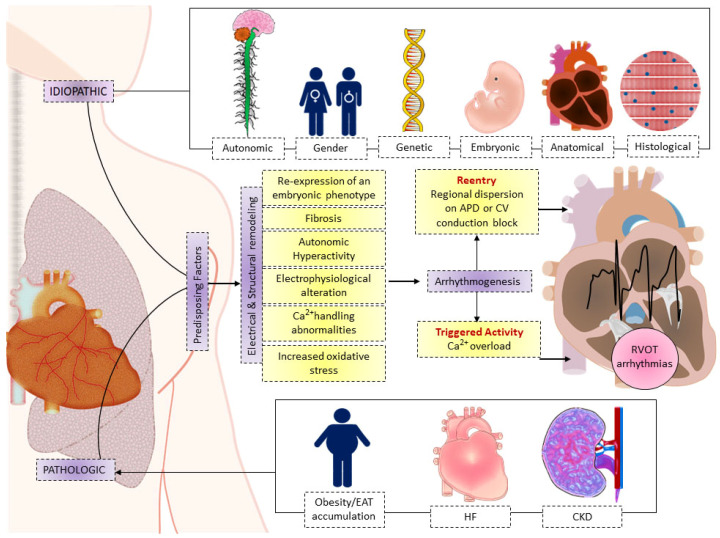
Schematic illustration of the intrinsic and extrinsic factors and possible mechanisms resulting in right ventricular outflow tract (RVOT) arrhythmogenesis due to electrical and structural remodeling. The idiopathic (autonomic, gender, genetic, embryonic, anatomical, and histological) and pathological conditions (obesity/epicardial adipose tissue (EAT) accumulation, heart failure (HF), and chronic kidney disease (CKD)) cause RVOT electrical and structural remodeling and are prone to induce triggered activity and reentry, resulting in RVOT arrhythmogenesis. APD, action potential duration; Ca^2+^, calcium; CV, conduction velocity.

**Table 1 ijms-24-11795-t001:** Genetic risk factors associated with right ventricular outflow tract arrhythmogenesis.

Associated Gene	Inheritance	Mechanism	Model	Reference
*ATP1A2, CACNA1C, PPP2R2C, PLCD3, GNAO1, Solute Carrier Family 6 (Transporter, Norepinephrine), Member 2(SLC6A2), CAMK2B, PIK3R2*	Down regulation	Myocardial intracellular Ca^2+^ regulation	Human	Hasdemir C. et al. [79]
*CAMKK2* and *ITPR3*	Up regulation			
*TBX3, BMP2, BMPR1B, MYH6, ANKRD23–39*	Down regulation	Cardiovascular functions	Human	Hasdemir C. et al. [79]
*RGS1*	Up regulation			
*Scn5a*	Heterozygous missense mutation	Na^+^ channel dysfunction	Mouse	Boukens BJ. et al. [22] Zhang Y. et al. [80] Martin CA. et al. [81] Pannone L. et al. [82] Chen Q. et al. [76]
			Human	
*Gja1*	Down regulation	Lower expression of gap junction	Mouse	Boukens BJ. et al. [22]
A1 ADO receptor (R296C)	Somatic mutation	Adenosine insensitive	Human	Cheung JW. et al. [83]
Inhibitory G protein Gαi_2_ (F200L)	Somatic mutation	Increase intracellular cAMP concentration Adenosine insensitive	Human	Cheung JW. et al. [83] Lerman BB. et al. [84]
TREK-1 (I267T)	Heterozygous point mutation	Stretch-activated K_2P_ K^+^ channel TREK-1	Human	Decher N. et al. [85]
Androgen receptor	Knockout	Myocardial intracellular Ca^2+^ regulation	Mouse	Tsai WC. et al. [74]
Stimulatory G protein alpha-subunit G_s_α (W234R)	Somatic mutation	Increase intracellular cAMP concentration	Human	Ip JE. et al. [86]

*ANKRD23–39*, Ankyrin Repeat Domain 23 and 39; *BMP2*, Bone Morphogenetic Protein 2; *BMPR1B*, Bone Morphogenetic Protein Receptor, Type IB; Ca^2+^, calcium; cAMP, cyclic adenosine monophosphate; K^+^, potassium; Na^+^, sodium; *PPP2R2C*, Protein Phosphatase 2, Regulatory Subunit B, Gamma Isoform; *TBX3*, T-box 3; TREK-1, TWIK-related potassium channel 1.

## Data Availability

No new data were created or analyzed in this study. Data sharing is not applicable to this article.

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
