# Peer review of "Electrical and Structural Insights into Right Ventricular Outflow Tract Arrhythmogenesis"

_ijms, 2023, doi:10.3390/ijms241411795_

Round 1

Reviewer 1 Report

A very valuable discussion on the right atrial outflow tract, which according to the authors is responsible for several types of fatal and overall understanding of malignant arrhythmias; idiopathic ventricular arrhythmia, Brugada syndrome, tosade de pointes, LQRS. and arrhythmogenic right ventricular cardiomyopathy (ARVC). They describe the morphology of the RVOT and some common characteristics of the aforementioned arrhythmias, and then histological changes with the consequent infiltration of fat and connective tissue, which increases the vulnerability of myocytes when the effects of the autonomic nervous system, sexual hormones and stress increase. Connective tissue and fat cells spread RVOT myocytes and lead to the mentioned malignant arrhythmias. Embryology and genetic expression already in embryonic development are then described, which is manifested by the different origin of myocytes of the right ventricle and the outflow tract of the right ventricle. The trigger for arrhythmias is also discussed, and catecholamines, which have the highest sensitivity in the induction of malignant arrhythmias, are mentioned as the most important. Then the influence of cellular electrophysiology in RVBOT arrhythmogenesis and the interplay of catecholamines as an inducing factor and intracellular cAMP and cations (L-Type and oscillatory calcium release from the sarcoplasmic reticulum) are described. This part of the publication emanates enormous knowledge and shows the latest findings that can change the perception of malignant arrhythmias both according to the origin, the mechanism of origin, similarities and differences, and especially the knowledge about the importance of calcium homeostasis. This publication enriches our knowledge about etiology, morphology, genetics, embryology, the need for homeostasis, the influence of sex, age and exciting factors. It can certainly bring significant progress in therapy, prevention and protection of patients with malignant cardiomyopathies originating from RVOT.

In the paper, the need for ARVC to be recognized in some families earlier in childhood than previously thought was not emphasized enough.

The attachment contains our contribution to that opinion presented in the following paper. If this remark can be accepted, perhaps this opinion should be included in this valuable work - (with minimal changes, without obligation)

Malčić I et al. Rigt ventricular arrhythmogenic cardiomyopathy - have we avoided a family tragedy by applying contemporary and treatment approach?

Liječ Vjesn 2016;138:339–344

We congratulate the authors and I think that the publication should definitely be published.

Author Response

Response to Reviewer #1

Thank you very much for your detailed comments. Those comments were very instructive, and very helpful to this manuscript. The responses to those comments are dictated below.

  1. Regarding the specific comment “In the paper, the need for ARVC to be recognized in some families earlier in childhood than previously thought was not emphasized enough. The attachment contains our contribution to that opinion presented in the following paper. If this remark can be accepted, perhaps this opinion should be included in this valuable work - (with minimal changes, without obligation) Malčić I et al. Rigt ventricular arrhythmogenic cardiomyopathy - have we avoided a family tragedy by applying contemporary and treatment approach? Liječ Vjesn 2016;138:339–344.”

-We appreciate your comment very much. According to your suggestion, we emphasized the importance of early diagnosis in the revised manuscript (page 13, line 16-19, red font) as follows “Inherited genetic channelopathy is a significant cause of sudden death in children and adolescents, such as ARVC, and family history and genetic screening help identify asymptomatic carriers and are essential for early diagnosis in RVOT tachycardia [Malčić et al. Liječ Vjesn 2016 and Zhou et al. Front Cardiovasc Med 2021].”

The above descriptions are the responses to your comments and suggestions.

Sincerely yours,

Yi-Jen Chen, MD, PhD

Reviewer 2 Report

The review paper titled 'Electrical and Structural Insights into Right Ventricular Outflow Tract Arrhythmogenesis' is attractive, especially from a clinical perspective. The article is well designed and written, but more from a physiological perspective and physiological changes in the right ventricular outflow tract that lead to ventricular arrhythmias, with very little molecular biology or described genetic changes that might be behind the development of ventricular arrhythmias. Apart from Table 1 and the general mention of genes that might play a role in the development of RVOT arrhythmogenesis, the molecular basis was not described. Therefore, this article would be more appropriate for a clinical or more physiologically oriented journal.

Authors should make some corrections before submitting the article to the appropriate journal:

1. The first sentence of the abstract and the first part of the second sentence are almost identical.

2. None of the four figures mentioned and their descriptions at the end of the article are illustrated, whereas Table 1 is included twice in the paper.

3. In vitro and in vivo terms should be written in italics.

N/A

Author Response

Response to Reviewer #2

Thank you very much for your detailed comments. Those comments were very instructive, and very helpful to this manuscript. The responses to those comments are dictated below.

  1. Regarding the specific comment “The first sentence of the abstract and the first part of the second sentence are almost identical.”

-Thank you for this comment very much. According to your suggestion, we rephrased and merged these two sentence in the revised abstract (page 2, line 2-5, red font) as follows ”The right ventricular outflow tract (RVOT) is the major origin of ventricular arrhythmias, including premature ventricular contractions, idiopathic ventricular arrhythmias, Brugada syndrome, torsade de pointes, long QT syndrome, and arrhythmogenic right ventricular cardiomyopathy.”

  1. Regarding the specific comment “None of the four figures mentioned and their descriptions at the end of the article are illustrated, whereas Table 1 is included twice in the paper.”

-We are sorry for this missing. According to your suggestions, we described the content of the figures in detail in the Figure legend and the revised manuscript (page 8, line 9-24 and page 9, line 1-3, page 10, line 12-17, red font). Besides, we deleted the duplicated Table 1 from the revised manuscript.

  1. Regarding the specific comment “In vitro and in vivo terms should be written in italics.”

-Thanks for this comment very much. According to your suggestions, we corrected the in vitro and in vivo terms in italics (page 10, line 13, red font).

The above descriptions are the responses to your comments and suggestions.

Sincerely yours,

Yi-Jen Chen, MD, PhD

Reviewer 3 Report

The research "Electrical and Structural Insights into Right Ventricular Outflow Tract Arrhythmogenesis" proposed by Yen Lu et al. focused on the right ventricular out-flow tract (RVOT) is a major site of origin for ventricular tachyarrhythmias such as idiopathic ventricular tachycardia (VT), ventricular arrhythmia with Brugada syndrome, and torsade de pointes. This review covers current knowledge on RVOT pathogenesis. The paper is intriguing, but there are a few things that may be improved.

Major Comments -

1) Can the authors explain the developmental basis for RVOT arrhythmias?

2) The article is missing all of the figures -1,2,3, and 4.

3) Can the authors highlight the areas that require improvement for RVOT diagnosis?

4) A more concise presentation of the conclusion section is required.

5) From the key words please remove “pathophysiology” and “electrophysiology” words.

TThe whole manuscript needs language editing in order to make it easier to read and follow. 

Author Response

Response to Reviewer #3

Thank you very much for your detailed comments. Those comments were very instructive, and very helpful to this manuscript. The responses to those comments are dictated below.

  1. Regarding the specific comment “Can the authors explain the developmental basis for RVOT arrhythmias?”

- We appreciated this comment very much. We have described the developmental basis for RVOT arrhythmias in part 2.2. “Embryonic Development Relates to RVOT Arrhythmogenesis” (page 5, line 13-23 and page 6, line 1,2) and described it in detail.

  1. Regarding the specific comment “The article is missing all of the figures -1,2,3, and 4.”

- Thank you for this comment very much. We are sorry for the missing all of the figures. We add on all of the figures in the revised manuscript.

  1. Regarding the specific comment “Can the authors highlight the areas that require improvement for RVOT diagnosis?”

- Thank you for this comment very much. We highlight the areas that require improvement for RVOT diagnosis in the revised manuscript (page 3, line 14-22, red font) as follows: “However, the intimate and complex anatomy of the outflow tract limits the predictive value of ECG criteria alone to distinguish RVOT tachycardia from SVT with left bundle branch block or ventricular tachycardia originating from areas closed to the RVOT, such as the left ventricular outflow tract tachycardia and the aortic sinus cusp, etc. For the limitation of ECG recognition for RVOT tachycardia, ECG characteristics to differentiate these arrhythmias from RVOT tachycardia have been reviewed ECG characteristics to differentiate these arrhythmias from RVOT tachycardia have been studied [Dixit et al. J Cardiovasc Electrophysiol. 2003; Anderson et al. Cir Arrhythym Electrophysiol; Mariani et al. Arrhythm Electrophysiol Rev. 2021]. Except for ECG, late potentials on the SAECG were present in 78% of the patients with ARVC/D but were not present in any patient with RVOT tachycardia, and ventricular tachycardia in ARVC does not terminate with adenosine [O’Donnell et al. Eur Heart J. 203]. Besides, we emphasize the importance of family history and genetic screening in early diagnosis of RVOT arrhythmias to identify asymptomatic carriers, and described in the revised manuscript (page 13, line 16-19, red font) as follows: “Inherited genetic channelopathy is a significant cause of sudden death in children and adolescents, such as ARVC, and family history and genetic screening may help identify asymptomatic carriers and are essential for early diagnosis in RVOT tachycardia [Malcic et al. Lijec Vjesn 2016 and Zhou et al. Front Cardiovasc Med 2021]”

  1. Regarding the specific comment “A more concise presentation of the conclusion section is required.”

-We appreciate this comment very much. We agreed with your comment and we rephrased our conclusion to make it more concise in the revised manuscript (page 19, line 7-11, red font) as follows ”The RVOT becomes highly arrhythmogenic because RVOT myocytes have unique electrical and Ca2+ handling properties that are known to cause arrhythmias. Multiple intrinsic and extrinsic factors are linked to RVOT arrhythmogenesis, which is easily induced in pathological states (Figure 4). As a result, tackling these causes and underlying diseases may provide therapeutic prospects for treating arrhythmias originating from the RVOT.”

  1. Regarding the specific comment “From the key words please remove “pathophysiology” and “electrophysiology” words.”.

-We appreciate this comment very much. According to you suggestion, we remove these two words from key words

The above descriptions are the responses to your comments and suggestions.

Sincerely yours,

Yi-Jen Chen, MD, PhD

Reviewer 4 Report

Dear authors,

The authors Lu et al aimed to present a concise review of the current state of knowledge of right ventricular (RV) outflow tract tachycardias. They structured the review in the two main paragraphs, namely predisposing factors of RV arrhythmogenesis, relating to anatomical, embryonic, electrophysiologic, hormonal etc. factors, and pathological conditions for RVOT arrhythmogenesis, relating to systemic conditions such as heart failure. With this, the review covers a broad range of facts predisposing to, generating or maintaining RVOT-tachycardia. There is a lack of current reviews about RVOT-VT, and this manuscript offers condensed knowledge of different aspects of this issue.

There are nevertheless several issues to comment on:

-          Several sources are cited sloppily. For example in the introduction, the authors cite reference #9 to explain that premature ventricular complexes arising from RVOT infrequently can trigger idiopathic ventricular fibrillation or polymorphic ventricular tachycardia. However, this reference shows that PMCs were originated in 4/7 patients with LQTS or BrS from the Purkinje system (other example: ref. 45). Thus, references should be checked for context.

-          The authors group Brugada syndrome, torsade de points, long QT syndrome, and arrhythmogenic RV cardiomyopathy (ARVC) all with the lable RVOT tachycardia (see introduction and later throughout the text). However, this mixes structural and electrical diseases being on different levels in one common final path which might occur in each of it (for example, tachycardia in ARVC originating from the RVOT due to tissue degeneration in the RVOT due to disease). Thus, the authors should make clear in the beginning whether they whish to define RVOT arrhythmias as a common final path of a plethora of diseases, or a singular entity. This will influence the content of the review further downstream, and will in some parts require re-structuring or re-phrasing.

-          For explaining cellular or electrophysiologic mechanisms, the authors mix studies gained from human or animal specimen, whole animal studies, whole organ studies and cell culture. This is comprehensible, as not each mechanism was studied in each kind of sample. However, the authors should always make transparent which model the respective content of the manuscript is based on.

-          The authors should consider including Rudic et al., JAHA 2016.

-          The figures are not showing.

The Quality of English Language might seldomly need some improvement.

Author Response

Response to Reviewer #4

Thank you very much for your detailed comments. Those comments were very instructive, and very helpful to this manuscript. The responses to those comments are dictated below.

  1. Regarding the specific comment “Several sources are cited sloppily. For example in the introduction, the authors cite reference #9 to explain that premature ventricular complexes arising from RVOT infrequently can trigger idiopathic ventricular fibrillation or polymorphic ventricular tachycardia. However, this reference shows that PMCs were originated in 4/7 patients with LQTS or BrS from the Purkinje system (other example: ref. 45). Thus, references should be checked for context”

- We appreciated this comment very much. We should precisely describe the content of the cited article. The result of the study done by Haïssaguerre et al. (reference #9) showed that ventricular fibrillation induced by triggering beats elicited from the RVOT, notably in Brugada syndrome, and from the Purkinje system, notably in long QT syndrome. We corrected the description in the revised manuscript (page 4, line 3-5, red font) as follows “Triggering beats from the RVOT have a crucial role in initiating ventricular fibrillation associated with Brugada syndrome, as confirmed by the elimination of triggers and malignant arrhythmias after ablation [Haïssaguerre et al. Circulation. 2003].” According to your suggestion, we have carefully checked all cited references in the revised manuscript.

  1. Regarding the specific comment “The authors group Brugada syndrome, torsade de points, long QT syndrome, and arrhythmogenic RV cardiomyopathy (ARVC) all with the lable RVOT tachycardia (see introduction and later throughout the text). However, this mixes structural and electrical diseases being on different levels in one common final path which might occur in each of it (for example, tachycardia in ARVC originating from the RVOT due to tissue degeneration in the RVOT due to disease). Thus, the authors should make clear in the beginning whether they whish to define RVOT arrhythmias as a common final path of a plethora of diseases, or a singular entity. This will influence the content of the review further downstream, and will in some parts require re-structuring or re-phrasing”

- Thank you for this comment very much. We agree with your comment that we have to make clear in the beginning define RVOT arrhythmias as premature ventricular contractions and ventricular tachyarrhythmias originate from the RVOT in patients with and without structural heart disease. According to your suggestion, we re-phrased that in the revised manuscript (page 3, line 2-9, red font) as follows “The right ventricular outflow tract (RVOT) tachycardia is a form of monomorphic or polymorphic ventricular tachycardia originating from the RVOT and accounts for the majority (80%) of outflow tract ventricular arrhythmias. The RVOT is a dominant site of origin of premature ventricular contractions and ventricular tachyarrhythmias that are often observed in patients without structural heart diseases, including idiopathic ventricular arrhythmias, Brugada syndrome, torsade de pointes, and long QT syndrome. The RVOT is also one of the origins of malignant ventricular tachycardia caused by structural heart disease, such as arrhythmogenic RV cardiomyopathy (ARVC) [Kamakura et al. Circulation 1998; Morita et al. J Cardiovasc Electrophysiol 2003; Tsai et al. Int J Cardiol 1997; Kim et al. J Am Coll Cardiol 2007].”

  1. Regarding the specific comment “For explaining cellular or electrophysiologic mechanisms, the authors mix studies gained from human or animal specimen, whole animal studies, whole organ studies and cell culture. This is comprehensible, as not each mechanism was studied in each kind of sample. However, the authors should always make transparent which model the respective content of the manuscript is based on.”

- Thank you very much for this comment. According to your suggestion, we identified the animal models used in the studies mentioned throughout the manuscript and presented them in red font.

  1. Regarding the specific comment “The authors should consider including Rudic et al., JAHA 2016.”

-We appreciate this comment very much. According to your suggestion, we described the content of Rudic’s article in the revised manuscript (page 7, line 20-23, red font) as follows: ” In patients with Brugada syndrome, there was a negative voltage gradient between the epicardium and endocardium of the RVOT with significantly prolonged activation time in the endocardium and the activation‐recovery interval in the epicardium, resulting in a larger area of ST‐segment elevation >2 mV and T‐wave inversions [Rudic et al. JAHA 2016].”

  1. Regarding the specific comment “The figures are not showing.”.

- Thank you for this comment very much. We are sorry for the missing all of the figures. We add on all of the figures in the revised manuscript.

The above descriptions are the responses to your comments and suggestions.

Sincerely yours,

Yi-Jen Chen, MD, PhD

Round 2

Reviewer 2 Report

The authors addressed all the issues raised and improved the article with figures which contributed to the quality of the article.

Reviewer 3 Report

Page 5 - Line 4 the word "healthy" is repeated. 

Reviewer 4 Report

During revision the authors have adressed each comment of the reviewer. The authors have re-written several paragraphs and added valuable information about cellular electrophysiology. Thereby the manuscript has gained in structure and comprehensibility.The figures underline the manuscript very well.